# Understanding the Role of Negative Emotions in Adult Learning and Achievement: A Social Functional Perspective

**DOI:** 10.3390/bs8020027

**Published:** 2018-02-20

**Authors:** Anna D. Rowe, Julie Fitness

**Affiliations:** 1Professional and Community Engagement, Macquarie University, North Ryde 2109, Australia; 2Psychology Department, Macquarie University, North Ryde 2109, Australia; julie.fitness@mq.edu.au

**Keywords:** negative emotions, achievement, higher education, perceptions, qualitative research, functional theory, prototype

## Abstract

The role of emotions in adult learning and achievement has received increasing attention in recent years. However, much of the emphasis has been on test anxiety, rather than the wider spectrum of negative emotions such as sadness, grief, boredom and anger. This paper reports findings of a qualitative study exploring the experience and functionality of negative emotions at university. Thirty-six academic staff and students from an Australian university were interviewed about emotional responses to a range of learning events. Data analysis was informed by a prototype approach to emotion research. Four categories of discrete negative emotions (anger, sadness, fear, boredom) were considered by teachers and students to be especially salient in learning, with self-conscious emotions (guilt, embarrassment, shame) mentioned by more students than staff. While negative emotions were frequently viewed as detrimental to motivation, performance and learning, they were also construed under some circumstances as beneficial. The findings are discussed in relation to the value of social functional approaches for a better understanding of the diverse roles of negative emotions in learning and achievement.

## 1. Introduction

The role of emotions in learning has received increasing empirical and theoretical attention in recent years [1,2,3]. However, much of the emphasis has been on test anxiety and (more recently) achievement emotions, that is, emotions directly related to achievement activities and outcomes [1,4]. Even so, a growing number of researchers also recognize that students experience a range of other emotions in academic settings including topic (i.e., subject area), epistemic and social emotions [5,6,7]. Epistemic emotions are experienced in response to the knowledge-generating qualities of cognitive activities [8] while social emotions arise in response to social concerns, e.g., status, power and attachment [9]. Discrete negative emotions found to be important in academic contexts include anxiety, fear, frustration, anger, boredom, sadness, shame, hopelessness, guilt and embarrassment [1,7,10,11]. Such emotions have been linked to assessment and testing [12,13], receiving grades and feedback on performance [7,14], teacher behavior [15], personal study [16] and satisfaction with the learning experience [11]. Understanding the impact of these and other negative emotions is important for informing developments in practice, as well as promoting strategies for managing such reactions, which can in turn lead to improved learning outcomes (for a review see [1,2]).

Evidence on the impact of negative emotions within academic environments suggests that they are detrimental to motivation, performance and learning in many situations, although findings are variable. For example, test anxiety, the most studied emotion in education, has been found to impact negatively on academic achievement [4] as well as motivate effort to avoid failure [13]. The impact of other negative emotions on learning and achievement is less well known, although literature is starting to emerge particularly on boredom [3,13,17,18,19]. Similarly, there is interest in the roles of fear in avoidance behavior and in prioritizing information processing under particular circumstances [20], and the role of anger in undermining motivation and task-irrelevant thinking [13]. In general, negative emotions are held to be detrimental to the pursuit of achievement goals, investment of effort, cognitive processes (such as attention and memory), motivation, self-regulation and self-efficacy [16,18,21,22,23]. Even so, there are some circumstances in which negative affect can potentially be adaptive; for example, in motivating students to attain goals and reduce error making or to recover from a negative performance evaluation [1,2,4,24]. The precise effects of emotions on academic achievement are thought to depend on the interactions among various mechanisms (e.g., achievement goals, cognitive resources, learning strategies), as well as requirements of the task being undertaken [2,3]. A limitation of much of the available empirical evidence is that it has been undertaken with school students and therefore remains to be tested with college learners. 

The overall aim of the current study was to further explore the perceived consequences of negative emotions through interviewing teachers and students in a University context. In particular, we sought to build on prior work on academic emotions [1,3,6,16] and to ascertain the extent to which students and teachers perceived negative emotions to have potentially useful functions in the context of learning and teaching.

### 1.1. Theoretical Perspectives

The guiding theoretical approach to this research is social-functionalist [25]. Social-functionalist approaches to emotions provide a useful framework for understanding why particular emotions are likely to be experienced in different contexts such as academic settings. This perspective conceives emotions as felt responses to appraisals, or interpretations, of events [26,27]. They are relatively short lasting and can be categorized into discrete emotion families such as fear, joy, sadness, and anger. Emotions are thought to differ from moods which are generally viewed as more diffuse affective states (e.g., feeling up or down) with a less identifiable cause [28]. Research on general mood states theoretically treats all negative and positive discrete emotions as functionally the same, and therefore does not recognize the diverse roles emotions play. Hence, as Pekrun and Perry, and others have observed, it is insufficient to simply differentiate between positive and negative states in educational contexts; the functional differences between diverse types of emotions need to be considered [3]. 

Discrete emotion categories are often termed basic emotions because they are thought to serve a primarily evolutionary function, underpinning coping strategies and adaptation [25,29,30]. Specifically, emotions are held to motivate potentially adaptive behaviors that enable individuals to resolve particular types of interpersonal and intrapersonal difficulties. The experience of emotion elicits and coordinates changes in cognition, physiology, behavior and subjective experience [29,31,32]. Anger, for example, is typically elicited in the context of perceived goal obstruction and motivates attempts to remove the obstruction. The experience of anger includes increased attention to goals and the actions of others, higher levels of physiological activity (e.g., increased heart rate, respiration) and potentially, behaviors such as aggression. Emotions thus motivate behaviors such as approach or withdrawal from the event or situation that has triggered the emotional reaction. Further, it is individuals’ perceptions (appraisals) of triggering events/situations that determine which emotions are experienced (sadness, anger, etc.), with around 70% of emotions predictable from the goal-relevant events that precede them [33]. 

Within educational contexts, negative emotions are thought to support tasks that require thinking within the boundaries of externally provided rules (accommodation), while positive emotions support tasks that depend on exploration and the creation of new ideas beyond given rules (assimilation) [2]. Piaget’s (1954) notions of assimilation and accommodation—complementary processes of adaptation through which awareness of the outside world is internalized—provide a useful theoretical lens for understanding emotions related to completing tasks at university [2]. However, this framework does not account for the specific functions of discrete emotions, or address how emotions resolve other issues such as interpersonal problems. 

Pekrun’s control-value theory of achievement addresses the functional mechanisms of discrete emotions, distinguishing emotions according to three dimensions: valence (positive and negative); the object focus (activity and outcome) and degree of activation (activating and deactivating) [3,6]. Negative activating emotions include anger, frustration, anxiety, and shame while deactivating emotions include boredom, sadness, disappointment and hopelessness. The theory posits that two types of appraisals—control and value—are particularly relevant to achievement emotions. That is, the extent to which students feel they have control over achievement activities and their outcomes, as well as the extent to which these activities and outcomes are perceived to be important. Our work has established that other appraisals are also important, particularly in relation to situations involving giving and receiving of feedback on assessments, e.g., deservingness, agency and expectedness [34]. While Pekrun and colleagues acknowledge the importance of social and epistemic emotions, the control-value theory focuses on academic achievement, with less emphasis on other areas of learning [1,8]. Further work is thus needed to develop more wholistic theories to account for the multiple types of emotions in educational contexts. Our study seeks to extend Pekrun’s work by exploring a broader range of negative emotions and their potential functions across a range of learning situations, and to compare the perspectives of teachers and students in their emotion knowledge and understanding. 

### 1.2. Emotion Knowledge (Prototypes and Schemas)

Prototype theory allows for the investigation of specific functions played by discrete emotions, and is compatible with the empirical findings of basic emotion researchers who conceptualize emotions as differing along a positive/negative dimension, as well as categories of discrete emotions (e.g., sadness, anger) [30]. Emotion prototypes are held to be mental representations of emotion concepts and categories, which are typically organized around the important elements (or exemplars) of a particular emotion. A set of defining features thus distinguish one emotion concept from other (sometimes similar) concepts—with separation often considered a matter of degree (i.e., the extent to which an emotion has characteristics deemed to be typical of that category) [35]. Prototypes are conceptualized along a hierarchy with superordinate (e.g., emotion), middle (negative emotion), and subordinate (e.g., sadness, fear, regret) levels. 

The findings of prototype studies (e.g., [36,37]) show that peoples’ ‘emotion knowledge’, that is, the way they understand the “expressions, feelings, and functions of discrete emotions” [30] (p. 44), has “considerable impact” on their perceptions, expectations and memories of emotion events [38] (p. 196). Thus, people draw on emotion prototypes to label and interpret their experiences of emotion, as well as to choose courses of action [39]. A student’s cognitive representations of anger (in response to perceptions of injustice over a grade) for example, will influence whether and how this emotion is expressed toward their teacher, and equally, how the teacher’s representation of anger will influence their behavior towards the student. Emotions schemas (prototypes) therefore play a critical role in shaping people’s perceptions, expectations, interpretations, memories and behaviors. 

Interestingly, there are an extensive number of studies on emotion knowledge in young school aged children but not for university/college students [40,41]. The prototype approach offers an alternative to examining scientific accounts of emotions in educational contexts, which while valuable, do not necessarily shed light on the underlying mechanisms of how people perceive, interpret, remember and act upon their emotions. Exploring student and faculty (i.e., lay persons’) theories of emotion, provides a fruitful avenue of investigation of this complex area, with the usefulness and applicability of prototype theory to the study of emotions and learning/achievement recently established [34]. 

### 1.3. Emotions in Academic Settings

The experiences of students and teachers (including the emotions experienced by each) impact on the quality of student learning, achievement outcomes and relationships [42,43]. While not an explicit focus in this paper, the nature of the student-teacher relationship differs from other types of interpersonal relationships (e.g., romantic; workplace) where a large amount of emotion research has taken place. It follows that the emotions experienced in academic contexts are likely to differ in response to different issues and therefore need to be considered. Firstly, the student/teacher relationship is hierarchical with unequal power structures/status, which is itself an important factor shaping how laypeople understand the way emotion episodes play out (e.g., emotion schemas/scripts) [44]. Given the structural power imbalances in university settings it might be expected negative emotions such as anger to be experienced by students because of perceptions of unfairness/injustice (over grades, feedback, unmet expectations etc.) or hopelessness, distress or depression over a perceived lack of control regarding academic outcomes [6,26,44,45]. 

Secondly, the supportive/nurturing element of teaching means that emotions such as anger, shame, guilt, boredom, hurt, hopelessness and anxiety might be experienced in response to a perceived lack of caregiving [15]. The importance of quality student-teacher relationships to students’ success at university is widely recognized [43], and understanding the interplay of social emotions experienced by students and faculty within educational settings is crucial, given interpersonal communication is a key trigger for emotional responses [9,46]. Thirdly, the achievement focus of higher education and potential for success and failure means that self-conscious emotions such as shame and guilt would likely be salient experiences among students, as well as anxiety, unhappiness, disappointment, hurt and sadness related to poor performance or negative feedback [6,24,47]. Fourth, the processes of learning and generating knowledge are likely to elicit epistemic emotions such as frustration, uncertainty, anxiety, confusion and boredom, which can impact on students’ use of cognitive and metacognitive learning strategies [8,48,49]. Finally, evidence suggests that some emotions (and their appraisals) can be domain specific, that is, are experienced in relation to specific discipline areas (e.g., math-related anxiety) [50]. 

The impact of emotional experiences on motivation and behavior within academic settings are variable. Negative activating emotions (e.g., hopelessness and boredom) are typically (although not always) associated with avoidance and therefore can undermine achievement motivation due to low-control appraisals (hopelessness) or perceived lack of incentives to perform academic activities (boredom) [3,18,19]. Negative activating emotions can lead to approach or avoidance behaviors depending on the underlying appraisals. For example, anger and anxiety are both negative activating emotions, but anger is approach-related—often triggered by appraisals of obstruction and unfairness [51], whereas anxiety is associated with avoidance of situations perceived to be threatening [4]. 

In summary, it was expected in the current study a range of achievement, epistemic and social emotions such as boredom, fear/anxiety, frustration, guilt and shame would be reported. Secondly, the perceived impact of such emotions would vary—in some cases prompting approach and in others avoidance behavior, which would have varying consequences for achievement and learning outcomes, in line with previous research and theory [1,3,6]. However, the overall approach to the research was exploratory, given that we still know little about the adaptive functions of emotions in learning and achievement, and the factors that mediate emotional responses in academic settings.

### 1.4. Qualitative Approaches to the Study of Emotions

Qualitative exploratory analysis is considered an effective approach for developing insights into the range and phenomenology of an individual’s emotion knowledge and the generation of hypotheses and theory which can inform the construction of quantitative measures (i.e., to draw generalizations and analyze more precise causes and effects) [1,16]. A qualitative approach was chosen for this research because of the need for exploratory research this area [1]. As Pekrun and Stephens note, such research is “necessary to judge the relative importance of different emotions as experienced by different students, and in different types of academic situations … [and] to generate more comprehensive conceptions of the contents and functions of student emotions, beyond hypotheses that can deductively be derived from existing theories” [1] (p. 296). Our aim was to obtain an in-depth understanding of student and teacher perspectives, given the extant focus on student perspectives [13,16]. Qualitative research can provide “in-depth contextualized understandings of human behavior and accounts of personal experience and meaning that may not be possible with quantitative methods” [52] (p. 29). The importance of investigating student and teacher perceptions is well established—both have views about the factors that enhance and hinder learning and such perspectives are valuable for understanding diverse factors that drive the learning process and its outcomes [53,54]. Importantly, it is also likely that students and teachers do not always understand their own or each other’s emotions in congruent ways, leading to the possibility of conflict over which emotions are more or less appropriate to be feeling, regulating, or expressing, in particular kinds of teaching and learning contexts.

### 1.5. Aims and Research Question

In line with the social functional approach discussed above, this paper focuses on students’ and teachers’ perceptions and understandings of negative emotions in a University context. Findings presented form part of a broader qualitative study exploring the experience and functionality of emotions at University, with analyses of positive emotions previously reported [10]. Specifically, the research investigates the following exploratory question: What are the features, functions and consequences, according to University students and teachers, of discrete negative emotions on learning and achievement (i.e., the impact on educational outcomes)?

## 2. Materials and Methods

### 2.1. Participants

Interviews were undertaken with 36 university staff (students = 21, faculty = 15) at an Australian university, with participants (female = 22, male = 14) located across a range of disciplines including business, science and humanities (Appendix A). The average age of students was 22 years. Age demographics were not collected for staff as it was not deemed relevant (it could be assumed Faculty were older). A decision was made to interview students and faculty as few emotion studies have reported on the perspectives of both groups. Additionally, we wanted to gauge the extent to which their views aligned. All participants were allocated a pseudonym.

### 2.2. Materials and Procedure

Participants were recruited via email, with invitations sent via various staff and student groups in keeping with a non-random (self-selecting) sampling approach, whereby participants volunteered to take part in the research, i.e., no particular individuals or groups were targeted. The diverse sample enabled the exploration of emotions and learning in a broader sense, rather than investigating potential demographic or disciplinary differences. Interviews were conducted on campus by two research assistants and participants received one movie ticket for their contribution. Semi-structured interview questions were developed by the authors following a review of the literature, and were supplemented with the results of a previous study [55]. Questions were intended to encourage participants’ recollections of their emotional responses (and in the case of Faculty, the emotional responses of students) to a range of learning situations and events under six broad themes: key emotions in learning; positive emotions; negative emotions; motivation; feedback and student/teacher relationships. While some specific areas were covered (e.g., feedback), in most cases the academic situations and events were not specified, as it was of interest to the researchers what types of situations/events would be reported.

Both groups were interviewed around the same theme areas, however questions focused on the student experience, not that of Faculty. There were some exceptions to this, e.g., a starter question asked students to describe a time when they felt bad about the learning experience, while faculty were asked to describe a time they felt bad about teaching. Generally, however, students were asked about their experiences of emotions in learning, while Faculty were asked about their perceptions of student emotional responses, and their role as teachers in facilitating and/or managing such responses (e.g., calming anxious students). This distinction was intended to allow for a comparison of perspectives in light of power differences between the two groups. Some faculty however, did speak of their own affective reactions in the course of the interviews. 

Starter questions for themes most relevant to this paper (e.g., excluding positive emotions) were as follows—these were designed to prompt rich accounts, with probing questions used if initial questions did not lead to developed responses:What emotions do you think are important in the learning process? (Participants were asked to give examples of when they felt bad about learning/teaching, reporting the emotions experienced and why it was a negative experience); What strategies do students use to manage negative emotions (e.g., felling stressed, upset) in order to successfully complete tasks/focus on their work? (Probing questions asked participants to comment on whether these were successful and what factors helped, e.g., peers, staff);What motivates students to learn? Do you think negative emotions are ever motivating?What emotions do you experience when you receive feedback? (Students) How do students feel about receiving feedback on their assignments? (Faculty) (Probing questions focused on identifying the impact this has on student motivation, if any, and strategies to mitigate negative emotional reactions triggered by feedback). 

Despite the small sample size, data saturation was reached and the sample was considered sufficient for an exploratory qualitative study [56]. All participants gave their informed consent for inclusion before being interviewed. Ethics approval for this research was granted by Macquarie University (ethical approval code: HE26SEP2008-D06071). 

### 2.3. Coding and Analysis

Coding of data was undertaken using QSR International NVivo 11 software (Doncaster, VIC, Australia) and informed by inductive and deductive methods of analysis. Predetermined coding categories were used to code and group emotions. This was to allow for an analysis of basic and discrete emotions, rather than broad negative affect. Coding (that is the classification and organization) of emotions adopted a combination of categorical (e.g., discrete categories such as anger, sadness) and dimensional (e.g., positive, negative) approaches to emotion measurement informed by a prototype approach [37,57]. The use of prototype approaches to emotion knowledge, concepts and scripts is well supported by empirical findings [36,37] and is compatible with dimensional and categorical approaches to emotion measurement as “it addresses both the contents of individual categories (e.g., the category of sadness episodes) and the hierarchical relations among categories (e.g., loneliness is a type of sadness, which itself is a type of negative emotion)” [58] (p. 186). 

Emotions were initially coded using participants’ own words, then as a second step, they were coded according to valence (positive/negative). Finally, they were grouped into basic emotion categories following previous research on emotion prototypes (largely drawing on Shaver et al.’s 1987 model) (Figure 1). Note in Figure 1, only select examples of discrete emotions are provided and disgust was removed due to its relative absence from the data. The allocation of discrete emotions to these broader dimensions was based on previous research, including Shaver and colleagues’ hierarchical organization of emotion knowledge [37], and the findings of basic emotion researchers such as Izard [32,59,60]. Across the data set, six negative ‘basic’ emotion dimensions were identified: anger, fear, sadness, disgust, boredom and self-conscious emotions (this category included some positive emotions as well as negative ones, i.e., guilt, shame, embarrassment and pride). Given the explorative nature of the research, having positive and negative emotions contained within one category was not considered problematic.

Remaining coding categories emerged from the data through elemental (e.g., descriptive) and thematic methods of coding and were revised over several cycles. The process of theme identification followed a similar procedure to that outlined by Braun and Clarke where the essence of each theme was identified and considered in relation to other codes and themes [61]. Parent or primary coding categories for the subset of data analyzed in the present paper were: social functioning (communication), cognition, motivation and drive, performance and productivity. Sub-categories of themes were coded hierarchically under these broad headings in accordance with their relationship to the parent category (e.g., complaining, assistance seeking were coded under ‘Social Functioning’; conscientiousness, exploration and curiosity, persistence were coded under ‘Productivity, Effort and Achievement Potential’ (Table 2). This follows the same procedure undertaken for our analysis of positive emotions [10]. Coding was initially conducted by the lead author and refined over several cycles, with reliability of coding categories verified through face to face meetings and email communication with other members of the research team [62]. For the study more broadly, data source triangulation (i.e., drawing on the voices of students and lecturers, the literature, and multiple theories), was used to enhance the validity of the research [63]. Triangulation of participants, for example, allowed for the comparison and crosschecking of consistency of information, as well as justification of coding themes, thereby enhancing the credibility of findings [63,64].

Matrix coding queries were generated in NVivo to cross reference parts of the data where emotions were reported in relation to their impact on learning and achievement. Interestingly, with the exception of motivation, interview questions did not explicitly ask participants for their thoughts about the impact of emotions on cognition, performance and other areas of functioning. Rather these were revealed in responses to other questions and identified through the coding process. Anger, boredom and fear were identified as key emotions, as measured by the higher frequency of mentions—others were cited less regularly by participants, suggesting these three emotions may play a more prominent role in enhancing and inhibiting learning. However, care must be taken in interpreting this finding as one of the interview questions specifically asked participants about managing stress (which might be associated with anxiety and related emotions in some people’s minds). Hence, the focus of this analysis is on the range of emotional experiences reported and their potential functions, rather than the importance (or not) of particular emotions.

## 3. Results

The research question aimed to identify perceived features and consequences (if any) of negative emotions on learning and achievement. It was anticipated that findings would shed light on specific functions served by negative emotions in educational contexts, thus extending previous theoretical work by emotion scholars. While the focus of the study was to identify the range of ideas and views reported, frequency data (generated through matrix queries) is also presented. 

### 3.1. Comparison of Student and Faculty Perceptions

As expected, anger, boredom and fear were reported frequently by both groups, although as mentioned earlier, care must be taken in interpreting this finding (Table 1). Interestingly, students also mention sadness and self-conscious emotions more than Faculty. This could be an artifact of the data, due in part to the focus on student emotions in the interview questions. Alternatively, the discrepancy could be related to the respective roles of each participant group. Given intense/powerful emotional experiences are remembered longer and recalled with greater vividness [65], it follows that the types of negative memories for each group might also differ, e.g., for teachers’, salient negative experiences could include dealing with angry students dissatisfied by grades/feedback as well as observing students bored in class, while for students it might be emotions tied to unmet achievement goals.

Participants reported that negative emotions can have either a positive or negative impact on different areas of functioning, suggesting a complex relationship between negative affect and learning (Figure 2). Interestingly, compared to students who generally viewed negative emotions as detrimental (with the exception of fear), teachers were more likely to emphasize the functional aspect of negative emotions. In particular, Faculty appeared to have a wider appreciation of the potential usefulness of emotions such as anger and frustration in motivating students to engage in a deeper way with their learning and complete assessment tasks. This was also reflected by the overall number of comments referring to the impact of emotions on learning and achievement, with almost double the number of coded remarks from teachers, despite the higher number of student participants (Table 2). Note that coding references are not a definitive measure but provide a general indication of the importance or prevalence of particular ideas/themes (as reflected by a higher number of coding references). The majority of student comments concerned increases and decreases in motivation, while Faculty comments ranged across a broader range of areas including behaviors (common ones being immobilization, complaining, assistance seeking, and increases in motivation/drive).

According to Faculty, the extent to which emotions promoted or hindered learning was mediated by several factors including gender, age and life experience (with mature age students perceived as being more able to cope with adversity), the intensity of emotions experienced, personality factors, and the nature of learning tasks. Emotion intensity was thought to be particularly important, with a “balance” needed between milder negative emotions (perceived to enhance learning) and more intense emotions (“too much stress and anxiety [is] detrimental to their [students] learning” Hilary, Accounting Faculty).
If they worry sometimes, at least it means they respect the difficulty of the task. But if it goes to an extreme and their anxiety is at a level where they can’t function, it becomes destructive.(Sheldon, Accounting Faculty)

Individual personality differences were considered important both in terms of learners’ being predisposed to react and behave in certain ways, as well as possessing the capacity to cope (or not) when experiencing negative emotions (i.e., individual resilience). Specifically, certain types of students were “stress resistant and [able to] … cope very well by their nature” (Benjamin, Business Faculty). As further explained by a faculty member:
There are people who just freeze and those who act. So the responses are different… Some of them would respond to it as if—it completely immobilizes them and they can’t function anymore and that is not a good way. And I imagine that those are the students that really need to consult with their tutors or their lecturers or would probably need to seek counseling or some support group that would enable them to actually manage that fear in a positive way… I don’t think we can just say fear is disabling.(Frances, Accounting Faculty)

“Anxiety and stress” were considered by some to be “an important part of being a high achiever … But [was] detrimental if it’s too extreme” (Hilary, Accounting Faculty). Benjamin noted that gender might play a role, suggesting that male students may “not be able to be willing to express their frustration and come and talk to you [the teacher]”; rather they were prone to aggressiveness, whereas females were more able to ask for help. Finally, the nature of the task was also important, e.g., some students find public speaking extremely fear provoking, more so than for other tasks, such as private study.

### 3.2. Cognitive and Social Functioning

Both participant groups perceived cognition to be impaired by negative emotions (especially fear and to a lesser extent, boredom and anger). These emotions reportedly contributed to difficulties in concentrating, focusing, paying attention to detail (with the exception of fear), absorbing and/or understanding information and instructions:
It made me unhappy and more stressed, because I really wanted to get some studying done but I found myself just being distracted by the emotion. So I’d find myself doing things like going to a tute [tutorial] for an hour, then the next day I wouldn’t have a clear understanding again.(Nadine, Science Student)

In terms of cognitive engagement, negative emotions were thought to close the mind off to ideas and connections with others (“some [students] will not respond and will just reject me again because I’m offering a hard solution”), as well as inhibit learning through freezing or immobilizing so “they [students] can’t function any more” (Frances, Accounting Faculty). Interestingly, only Faculty commented on the immobilization function, suggesting students may not be aware, or have the language to articulate this process yet. Typical comments were similar to those expressed by Benjamin, a business/marketing lecturer, who observed that with some forms of anxiety “you start to feel emotionally switched off, and you’re no longer open to engaging with the material.”

Social functioning was viewed as being both enhanced and impaired by negative emotions. Assistance seeking behaviors were prompted by feelings of frustration, fear and confusion. For example, students approached teaching staff, peers, friends and family to clarify issues they were unsure about or to ask for help, including application for special consideration in times of hardship. Special consideration is the formal name of the process at the university, whereby students can request compassion for their study/performance during periods of hardship, illness etc.
I’ll never leave something unresolved if I’m not sure with what they’re [teachers] trying to say or if I’m really unhappy with what they’ve written.(Luke, Science Student)

At the same time, fear and anger were also thought to impair assistance seeking behavior and communication more broadly. For example, Faculty considered that students might feel too “intimidated to come to the teacher” while angry students approached teachers directly to complain or pressure them to change “unfair” outcomes such as poor grades. While such behaviors could be considered adaptive, they were not perceived by lecturers to be cooperative or helpful to the learning process:
There are different ways of dealing with frustration and it’s not always channeled the right way, as in asking for help or assistance.(Benjamin, Business Faculty)

Embarrassment could also hinder assistance seeking through avoidance: “so you sort of felt embarrassed to ask a few questions because you didn’t know how they [the teachers] were going to react” (Wendy, Chiropractic Student).

### 3.3. Motivation, Productivity and Performance

According to both participant groups, negative emotions had the capacity to increase as well as decrease motivation and direct effort into study/task completion, resulting in varied levels of productivity and achievement. Like cognitive engagement, there was a general sentiment that “a certain amount of stress is good; like not to be too relaxed. [That] You need to be a little bit stressed to really achieve perhaps your potential” (Hilary, Accounting Faculty). Similarly,
The fear of failure is a reality and therefore those sorts of negative emotions are usually present and can be motivating and I think that we have to recognize that while the fear of failure is not what you want to be a dominating sort of emotional motivator, it’s part of the package.(Cameron, Business Faculty)

The experience of frustration encouraged exploration and curiosity, suggesting epistemic emotions may have an important role to play here:
*…frustration is part of curiosity, in a sense, that I want to know more and I can’t. You have to sort of shut against a brick wall for a while before it falls down. There’s a degree of frustration that’s quite healthy because it fuels your own desire to push forward. You know you’ve got something to push against*.(Michelle, English Faculty)

Frustration, in particular, was held to motivate students to understand concepts, as well as challenge their assumptions: “If they’re frustrated because they don’t understand it, that motivates them to take the steps needed to overcome their lack of understanding” (Sheldon, Accounting Faculty). Increased persistence, especially on undesirable tasks, was also associated with negative emotions such as dislike, uncertainty and unhappiness. Thus, they appear to play a role in promoting diligence and the commitment to follow through, especially on difficult or unpleasant tasks. Despite the perceived detrimental impact of negative emotions on attention broadly, interestingly fear was associated with increased conscientiousness, suggesting a role in promoting quality outcomes:
[fear] makes sure you cross your Ts and dot your Is. It ensures that you cover every little detail because you’re so terrified that you’re going to, if it’s a public speaking thing, stuff up, get it wrong; make a mistake in an exam, miss an issue.(Hilary, Accounting Faculty)

Despite some of the benefits for motivation, on the flip side negative emotions (particularly boredom and fear) could equally promote avoidance behaviors such as withdrawing from courses, swapping classes, procrastinating, sleeping in, and the general urge to “run away.”
I’ve seen several times in three years … people withdrawing because they felt so stressed that they just couldn’t cope with coming in class, fulfilling their work and everything.(Genevieve, French Faculty)

Boredom was viewed as strongly hindering motivation and cognition, and one of the more challenging emotions to manage as a teacher:
If a student’s bored, you know, in the long run, no matter what you do as a teacher, no matter how much you try and engage them in the most innovative creative ways you can possibly think of, if they don’t want to be there… I don’t think there’s anything you can do.(Angelina, Cultural Studies Faculty)

Only three participants held the view that boredom could be motivating and productive in some circumstances. Negative emotions were also associated with encouraging surface learning approaches, such as rote learning materials the day before an exam, and other strategies considered undesirable by teaching staff. Sadness appears to play a role in both inhibiting and promoting motivation, particularly around experiences of failure and/or poor performance evaluations, but little else. Negative self-conscious emotions (shame, grief and embarrassment) were seldom mentioned in terms of their impact on learning or achievement, but were dominant elsewhere in the data, particularly in relation to the antecedents (appraisals) of emotions—notably experiences of social exclusion/ostracism. Where reported, feelings of guilt and inadequacy were linked to an increased drive to succeed and/or investment of effort:
…it really did impact on my self-esteem. It really did just make me feel quite inadequate as a student … being told that I’m not smart enough to do something, I now have this constant drive to want to be just smart enough to do something.(Kevin, International Studies Student)

Finally, a strong theme of “waste” was evident in participant’s accounts of the lower productivity and achievement resulting from negative emotions.
Uncertainty, fear, regret; like maybe I thought I was better able to do this work than I am so maybe I’ve wasted my time and my money and her [the teachers] time. Yeah, doubt, total doubt. Fear, like oh my gosh, this is where I thought my career was going, this is my study, and this is the postgrad so it’s kind of the culmination of my undergrad and study, and all of these years of—so has that all been wasteful.(Phoebe, Arts Student)

In summary, a range of impacts were reported by participants, which provide clues as to the potential functions of particular negative emotions in learning and achievement at university.

## 4. Discussion

### 4.1. The Functions of Negative Emotions

Patterns were observed in the data in relation to the effects of anger, fear, sadness and boredom across four main areas of functioning (Table 1). Unlike positive emotions, which seem to have a mostly positive impact on learning (e.g., [10,66,67]), participants reported that negative emotions can have either a positive or negative impact. This suggests a more complex relationship between negative affect and learning, a view supported by existing literature (e.g., [1]). Interestingly, adaptive functions were recognized more by Faculty, with students focusing on motivational consequences. The view that a certain level of negative emotion was needed to enhance engagement resonates with the Yerkes-Dodson Law that proposes a certain degree of stress or arousal can actually improve performance [68]. Anger and fear are clearly perceived to be both enablers and inhibitors of learning and achievement, impacting across all areas of functioning. Anger and anxiety (coded here as part of Fear) are both considered activating emotions [3], however they likely serve different functions, with anger generally approach-related (e.g., students’ complaining to lecturers about grades) while anxiety is avoidant (e.g., students’ not attending an exam because of a fear of failing). Pekrun and Perry note the motivational effects of negative activating emotions such as anger and fear are complex, as the outcomes can be so variable [3]. Our findings are in line with previous theory and research that has found anger functions primarily to remove obstacles to goal achievement, often through securing a better outcome by forcing a change in another person’s behavior [69]. Fear serves to address threats, by removing oneself from a perceived danger, supporting previous scholarship which shows fear is elicited in response to appraisals of immediate and specific threats, and narrows thoughts and actions [27,67]. The research also supports existing work on anxiety, which has been found to impair performance in testing situations [14]. The key to better understanding the nuanced effects of these emotions may be to look more closely at the underlying appraisals of each [7]. 

In the present data set, frustration at not understanding a concept or feeling dissatisfied with the current state of knowledge appeared to be associated with increased motivation, while anger or frustration attributed to teachers and/or the curriculum was linked with impaired motivation in participants’ minds. This may reflect personality variables, but it is interesting that frustration directed toward external factors (e.g., teachers, curricula) appear (at least in this instance) to be more detrimental to motivation than when directed at self. It points to potential differences in the role of frustration as an epistemic vs. an achievement emotion. Emotions can share affective properties with other categories of emotion, but differ in terms of their object focus [8]. Frustration could be considered an epistemic emotion if the focus is on cognitive incongruity resulting from an unsolved problem, or an achievement emotion if the focus is on the curriculum or teacher as a perceived obstacle to academic success. In both cases frustration is an activating emotion which could in some circumstances promote strategy use (i.e., as an epistemic emotion) or persistence (as an achievement emotion), and is consistent with other evidence that unpleasant emotions can promote learning and do not always require remediation [70,71].

The finding that boredom hindered motivation and cognition, supports recent research suggesting boredom has a uniformly negative effect on learning and achievement outcomes [19]. However, recent perspectives suggest boredom may play a role in encouraging the pursuit of alternative goals and experiences, by signaling to a person that it is time to try something new [17]. Mixed findings in the literature could be due in part, to a lack of clarity and agreement around the arousal levels of boredom—like frustration, boredom is considered a deactivating emotion [3], however recent research suggests there may be subtypes of boredom with differing levels of arousal and valence [72]. Another factor may be perceptions of agency. In the present study, boredom was associated primarily with two external factors—the curriculum (content) and its delivery (i.e., teaching style), with Faculty investing considerable effort to minimize boredom and its effects (e.g., disengagement, distraction). This suggests appraisals of agency are potentially an important antecedent to experiences of boredom, in addition to value and control which have already been theoretically and empirically established [18]. Further work is needed on this under researched emotion to determine more precisely its antecedents and impact.

All mentions of sadness were in relation to its intrapersonal effects, an area of research that remains largely untested [73]. There was insufficient data to draw any other conclusions from our data and this emotion requires further investigation in educational contexts. In the scholarly literature, sadness is a deactivating emotion elicited by appraisals of irreparable loss and therefore likely has implications for how students cope in the face of adversity, e.g., persistence, dropping out [27]. It may also play a role in helping students make sense of past failures in order to prevent future ones [74].

It is not clear why negative self-conscious emotions were not as strongly articulated as having an effect on learning, but were reported more in relation to emotional triggers. Existing literature on self-conscious emotions suggests they play an important role in achievement [47], so it could be that this is an artifact of the data given the majority of comments coded for negative self-conscious emotions were made by students (and students do not appear to recognize the adaptive benefits of emotions to the same extent as teachers). Future research would benefit from empirically investigating different types of shame, embarrassment etc. in tertiary settings, along with the self-evaluations that underpin such emotions, in order to better understand their variable outcomes on achievement and learning. Such findings could be used to inform instructional strategies to prevent and manage shame and related responses [75], which may include developing students emotion knowledge.

### 4.2. Mediating Factors

Several mediating factors were identified by Faculty as affecting the direction of impact (promoting or inhibiting) on learning and achievement following the experience of negative emotions. These were mostly mentioned in relation to fear, although it should be noted that the overall number of comments was small and should not be over emphasized. Demographic variables of age, gender and personality attributes were deemed to influence how people react, as well as hinder their ability to cope with and/or regulate negative feelings. Intensely felt emotions were thought be more detrimental, perhaps because they are harder to manage and regulate. Typically, (although not always) emotion regulation strives to increase positive emotions, and decrease negative ones [1] and a person’s capacity to mentalize will impact on their ability to regulate emotions, pointing again to the importance of individual differences [76]. While the nature of academic tasks was also mentioned, this received less emphasis than personal variables, which is surprising as task demands are emphasized in the literature as a major contributing factor to emotions experienced in educational settings (e.g., [2,6]).

It is beyond the scope of the present paper to discuss all these mediating factors in depth, however a couple are worth a mention. Gender differences in experiences of affect for example, have been reported in relation to domain specific areas such as mathematics, with adolescent girls reporting significantly more anxiety than boys [77]. Benjamin’s account of gender differences in dealing with frustration is suggestive of gender differences in emotion knowledge, that is, how female and male students understand frustration. In both cases described, the same emotion (frustration) is experienced, but different approach behaviors are enacted that are likely to lead to very different outcomes. Personality traits are another area worthy of further investigation. While they have been linked to academic performance and achievement some findings are mixed. For example, one study found neuroticism was negatively associated with academic achievement [78] while another reported it was not a strong predictor of GPA [79]. It could be that there is an inverted relationship, which means some amount of neuroticism may be beneficial but too much detrimental. This and other trait emotions are deserving of further follow up. In our study, while some faculty commented on personality characteristics of students, few mentioned trait emotions. One lecturer observed that chronic predispositions to experience particular emotions such as anxiety were problematic and required professional treatment. While not examined here, individual differences relating to gender, culture etc. are important as it is recognized that emotions are not necessarily similarly experienced or expressed across groups [58,80].

The dominant voice of teachers’ views of the impact of emotions on learning, as well as their wider appreciation of the benefits of negative affect, could be attributed to their age, in that they have a longer history of experiences with emotions in learning. Further, older adults are generally better able to differentiate positive and negative sources of emotion knowledge [81]. Given emotion knowledge “operates as a major (though not necessarily the only) factor in emotion regulation” [30] (p. 45) it could be assumed that that participants with a stronger emotion knowledge are better able to regulate their emotions, and therefore more readily see the benefits of negative affect. Such an assumption is supported by a recent study which found that while chronic negative affect (unhappiness) over the duration of a students’ degree program was detrimental to performance, the academic success of ‘happy’ students’ seems to originate from their ability to adaptively manage the motivational benefits of bouts of heightened negative affect [82]. 

### 4.3. Limitations

Findings reported in the present study are of perceptions of emotions, not induced or observed emotions. They provide a snapshot in time of the experiences of students and academics at one university, and cannot be generalized beyond that context. However, they can inform larger studies investigating more precise effects and causes (e.g., systematic measures of educational outcomes), as well as contribute to development of teaching strategies aimed at harnessing emotions to promote rather than hinder learning, and teaching students how to recognize and regulate negative affect. The sample was comprised of students and academics from diverse disciplines, which may affect their understanding of emotions, and therefore potentially influence the data. Other limitations include the general confines of using interviews as a data collection method (e.g., social desirability effects, memory bias, lack of openness on the part of participants) and of using interviews in emotion research (e.g., the potential to under report less intense emotion experiences over intense ones, willingness to disclose emotions) [31]. This is important as subtle affective cues can impact on cognition, i.e., it does not need to be an enduring, intensively felt emotion to have an influence on achievement, motivation or behavior [2]. Finally, we acknowledge the limitations of basic emotions as a construct (e.g., [83]) and of coding data in this way (particularly in light of debate around the status of some emotion categories, e.g., whether confusion should be defined as an emotion within its own right, as opposed to a sub-category of other more ‘basic’ emotions) [84]. For the purposes of this exploratory study, a prototype approach was deemed an effective and simple way of coding and analyzing responses in order to make meaningful inferences.

### 4.4. Implications for Practice and Future Research

There are practical lessons to be learnt from this data on how to both prevent and manage the fallout of negative emotions. A number of strategies were offered by participants to reduce the onset of negative emotions, most of which fell within the responsibility of teaching staff. They included the use of humor and other teaching methods to foster interest in the classroom (particularly when the content was considered ‘boring’), showing care and concern, encouraging students to form connections with peers and seek assistance in times of stress, providing frequent and timely feedback to reduce anxiety and confusion, being available to students, providing clear instructions and dispersing assessment tasks throughout semester to prevent student overload. Students’ ability to cope with and regulate negative emotions is also paramount, and reflective practice, cognitive reappraisal and other pedagogical strategies may offer useful approaches to promoting self-regulation [85,86].

There are a number of areas in need of further investigation. In addition to those already mentioned, the relationship between different categories of emotion (achievement, social, epistemic etc.) and co-occurring emotions (such as frustration/curiosity and fear/excitement in the present study) warrants additional research, as these can be overlooked in experimental paradigms of basic emotions which tend to focus study on the discrete boundaries that differentiate one emotion from another [83]. The antecedents of negative emotions in learning (both in terms of the types of events/situations and appraisals); power dynamics and discrete negative emotions within the context of student-teacher relationships, and more theorizing and empirical work around academic emotions not related to achievement (i.e., social and epistemic emotions) is also needed. Finally, future quantitative research needs to empirically examine our study’s assumptions, as well as building on existing work testing variations of emotional experience in educational contexts by gender, class, culture and other demographic variables to allow a richer contextual understanding of the experience of emotions in education (e.g., [87]). 

## 5. Conclusions

The research presented is a contribution for advancing the field of emotion research in learning and teaching. We have reported on emotions identified by students and faculty as being important to learning and achievement in higher education, as well as explored the meaning of those emotions across a range of learning situations. This offers an alternative perspective to scientific accounts of the role of academic emotions, with lay peoples’ knowledge about the nature and course of emotions such as anger, sadness, fear and boredom a valuable source of data about the functions these emotions play in educational contexts. Overall findings reveal that negative affect is perceived to both enhance and hinder learning, supporting previous empirical and theoretical work. Given the complexity of interactions between emotions and variables such as task requirements, interpersonal relationships, achievement goals and cognitive resources, and the lack of agreement over the precise mechanisms by which some emotions are able to bring about very different effects, perhaps researchers and theorists should modify the types of questions being asked. As Fielder and Beier observe, rather than asking for a “simplified, one-sided answer to the question of whether achievement and motivation profit from positive or negative mood”, perhaps we should ask what kinds of achievement, learning and motivation are enhanced by negative emotions? [2] (p. 51). Further research is needed to uncover the nuances of these interactions to better understand emotions, which have the potential to both enhance and inhibit learning, and add knowledge about the experiences of co-occurring emotions in the learning environment. 

## Figures and Tables

**Figure 1 behavsci-08-00027-f001:**
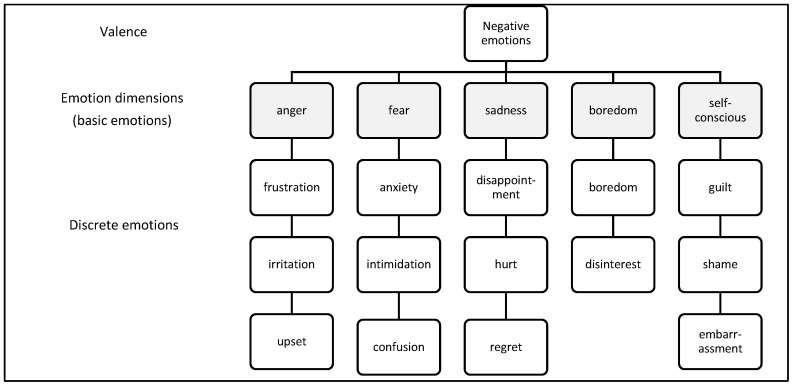
Negative emotion ‘family’ dimensions based on prototype analysis.

**Figure 2 behavsci-08-00027-f002:**
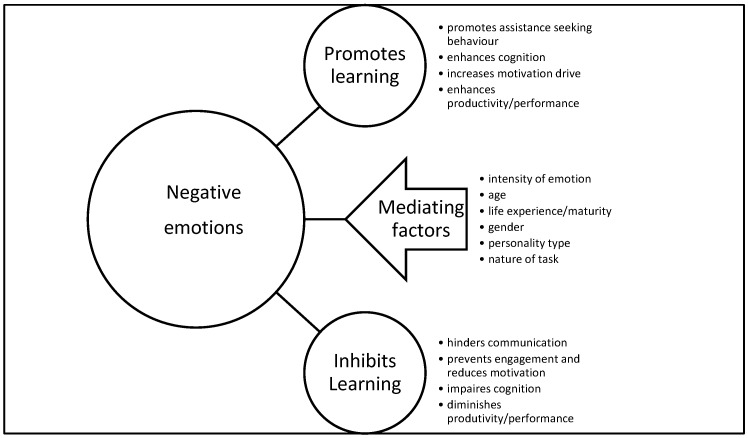
Perceived impact of negative emotions on learning.

**Table 1 behavsci-08-00027-t001:** Percentage of participants who reported discrete emotions contained within basic negative emotion categories.

	Student (%)	Faculty (%)
Anger	95	94
Boredom	81	75
Fear	100	94
Sadness	81	44
Self-conscious	81	56

**Table 2 behavsci-08-00027-t002:** Student and Faculty perceptions of the impact of negative emotions on learning as measured by number of coding references.

Perceived Impact	Students	Faculty
Promotes learning (generally)	0	1
Enhances cognition	0	4
Enhances social functioning (assistance seeking)	1	8
Increases motivation and drive	11	12
Increases productivity, effort and achievement potential	3	5
Heightens conscientiousness	0	2
Promotes exploration and curiosity	1	5
Promotes persistence	3	2
Inhibits learning (generally)	2	8
Lowers performance and productivity (wasted time)	2	2
Impairs social functioning (communication)	1	0
Complaining/trying to change unfair outcome	1	10
Prevents assistance seeking	0	3
Reluctance to provide feedback	0	1
Impairs cognition	6	8
Closes off mind to ideas or people	1	1
Immobilizes	0	10
Loss of motivation, avoidance and procrastination	16	5

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
