# Peer review of "Understanding the Role of Negative Emotions in Adult Learning and Achievement: A Social Functional Perspective"

_behavsci, 2018, doi:10.3390/bs8020027_

Round 1

Reviewer 1 Report

This revision demonstrates extensive work on the Authors behalf to address reviewers’ comments and concerns. Incorporating this feedback has certainly improved the manuscripts theoretical justification and methodological approach to their now clear research question. Below are some final comments.

Introduction Questions/Comments:

Authors are asked to earlier on define topic/discreet, epistemic, and social emotions and to be more consistent in the labeling and classification of emotions throughout. The Authors incorporate a new section discussing student/teacher relationships. I assume this was in an effort to justify their inclusion of Faculty participants. However, as written, it comes across that this will be an area of investigation that your study does not examine. There are several places where “causes” of emotions are still mentioned yet not the intent of the study.

Method Questions/Comments:

Authors are asked include their ICC’s when reporting their coding reliability procedures especially given the qualitative nature of the study.

Results Questions/Comments:

“Although we acknowledge that strategies used to manage negative emotions and their relationships with teachers are important there is insufficient space in the Results to examine these areas in more detail.” – Please make a clear statement of what your study will and will not discuss and why.

Authors are also asked not to explain results in the Results section but wait until the Discussion section to interpret their findings.

Discussion & Limitations Questions/Comments:

“The study explores a range of learning/academic outcomes. As the analysis was exploratory we did not measure each outcome systematically as the overall number of coding references would be small and therefore not statistically meaningful. Rather the focus is on the range of views/perspectives.” Add as limitation

Author Response

The authors wish to thank Reviewer 1 for their feedback and suggestions. We have made minor revisions to the paper as requested and our response is below.

Introduction: We have amended the Introductory paragraph to include definitions of the different types of emotions as requested by Reviewer 1.

The section on student/teacher relationships has been amended to soften the implication that this is an area of investigation in the study (which subsequently isn’t investigated).  

All mentions of “causes” of emotions in relation to the aims of the study have been removed.

Method Questions/Comments: Reviewer 1 requested that we include ICC’s when reporting coding reliability procedures. We concur with Reviewer 1 that demonstrating reliability is important, however ICC’s are a more appropriate measure of reliability for quantitative data. Reliability in our study (as articulated in the paper) was determined via an iterative process through collaborative discussions. To allay Reviewer 1’s concerns, and to strengthen the credibility of our findings we have addressed the area of reliability more broadly for the study in terms of data triangulation, supported by relevant literature.

Results Questions/Comments: Reviewer 1’s request to make a clear statement of what the study will and will not discuss and why was made in reference to an excerpt from the paper which was removed in the previous version. As this appears to be no longer applicable we have not actioned the request.

We have removed most interpretation from the Results (and into the Discussion), with the exception of one reference to the literature which is important to our justification regarding the analysis.

Discussion & Limitations Questions/Comments: the author’s understand Reviewer 1’s perspective, however our view is that the focus on the range of responses, rather than systematic measurement of particular outcomes, is a feature of this kind of study, rather than a limitation. Hence, we have not actioned this request.

Again, we would like to thank the reviewers for their very constructive feedback.

Kind Regards

Anna and Julie

Reviewer 2 Report

The authors have submitted a substantial revision of the paper. They have addressed my concerns that I addressed in the first review round satisfactory. The structure of the argumentation has improved and the used method is clearer now. I do believe that the paper makes a significant contribution to the field.

Just a very small remark: I am not sure if it is adequate to use the term "random sampling approach" (2. Materials and Methods) if a qualitative approach is used. In the end, the authors ended up with a "non-random-sample" as the participation was voluntary. Thus, it might be better to use another word. "Random-sampling" is normally a term that comes with quantitative approaches and confuses at bit in relation with the present study .

Author Response

The authors wish to thank Reviewer 2 for their feedback and suggestions. We have made minor revisions to the paper as requested and our response is below.

In response to Reviewer 2’s recommendation to use an alternative term to random sampling approach we have replaced the term ‘random sampling approach’ with ‘non-random sample’, specifically ‘self-selection sampling’ where by participants volunteered to take part in the research. We thank Reviewer 2 for bringing this issue to our attention.

Kind Regards

Anna and Julie